# The Prognostic Utility of the Metastatic Lymph Node Ratio and the Number of Regional Lymph Nodes Removed from Patients with Small Bowel Adenocarcinomas

**DOI:** 10.3390/medicina59081472

**Published:** 2023-08-16

**Authors:** Dincer Aydin, Umut Kefeli, Melike Ozcelik, Gokmen Umut Erdem, Mehmet Ali Sendur, Mahmut Emre Yildirim, Basak Bala Oven, Ahmet Bilici, Mahmut Gumus

**Affiliations:** 1Department of Medical Oncology, University of Health Sciences, Derince Training and Research Hospital, Kocaeli 41900, Turkey; 2Department of Medical Oncology, Faculty of Medicine, Kocaeli University, Kocaeli 41100, Turkey; umut.kefeli@kocaeli.edu.tr; 3Department of Medical Oncology, University of Health Sciences, Umraniye Training and Research Hospital, Istanbul 34764, Turkey; melike.ozcelik@sbu.edu.tr; 4Department of Medical Oncology, University of Health Sciences, Cam and Sakura City Hospital, Istanbul 34480, Turkey; gokmenumut.erdem@sbu.edu.tr; 5Department of Medical Oncology, University of Health Sciences, Ankara City Hospital, Ankara 06800, Turkey; mansendur@ybu.edu.tr; 6Department of Medical Oncology, University of Health Sciences, Kartal Dr. Lutfi Kirdar City Hospital, Istanbul 34865, Turkey; emremahmutyildirim@gmail.com; 7Department of Medical Oncology, Faculty of Medicine, Yeditepe University, Istanbul 34752, Turkey; basak.oven@yeditepe.edu.tr; 8Department of Medical Oncology, Faculty of Medicine, Medipol University, Istanbul 34214, Turkey; abilici@medipol.edu.tr; 9Department of Medical Oncology, Faculty of Medicine, Medeniyet University, Istanbul 34722, Turkey; mahmut.gumus@medeniyet.edu.tr

**Keywords:** lymph node metastasis, metastatic lymph node ratio, prognosis, small bowel adenocarcinoma

## Abstract

*Background and Objectives*: Small bowel adenocarcinomas (SBAs) are rare tumors of the gastrointestinal system. Lymph node metastasis in patients with curatively resected SBAs is associated with poor prognosis. In this study, we determined the prognostic utility of the number of removed lymph nodes and the metastatic lymph node ratio (the N ratio). *Materials and Methods*: The data of 97 patients who underwent curative SBA resection in nine hospitals of Turkey were retrospectively evaluated. Univariate and multivariate analyses of potentially prognostic factors including the N ratio and the numbers of regional lymph nodes removed were evaluated. *Results*: Univariate analysis showed that perineural and vascular invasion, metastatic lymph nodes, advanced TNM stage, and a high N ratio were significant predictors of poor survival. Multivariate analysis revealed that the N ratio was a significant independent predictor of disease-specific survival (DSS). The group with the lowest N ratio exhibited the longest disease-free survival (DFS) and DSS; these decreased significantly as the N ratio increased (both, *p* < 0.001). There was no significant difference in either DFS or DSS between groups with low and high numbers of dissected lymph nodes (i.e., <13 and ≥13) (both, *p* = 0.075). *Conclusions*: We found that the N ratio was independently prognostic of DSS in patients with radically resected SBAs. The N ratio is a convenient and accurate measure of the severity of lymph node metastasis.

## 1. Introduction

Malignant tumors of the small intestine are infrequent. Only approximately 3% of all gastrointestinal tumors originate in the small intestine despite its large surface area and length [1]. The mean age at diagnosis is between 50 and 70 years, and incidence is similar between men and women [2]. Adenocarcinomas, one of the four most common histological tumor types, are responsible for approximately 30–40% of all small bowel tumors [3]; the other most common tumor types are neuroendocrine tumors, lymphomas, and sarcomas. Small bowel adenocarcinomas (SBAs) most commonly arise in the duodenum (57–65%) and decrease in frequency more distally [4]. Given the nonspecificity of symptoms, such tumors are diagnosed at more advanced stages than colorectal cancers; 60% of patients are of stage 3–4 at diagnosis, compromising prognosis [5,6]. For most malignancies, surgical resection is the principal primary treatment for local (stage I–III) SBAs. Dissection of the primary tumor with *en bloc* lymph node removal is the favored surgical approach [7,8]. Lymph node metastasis status is very important in prognostic terms [9,10,11]. The pathological lymph node (pN) classification is prognostic and therefore used as a simple method for accurate staging [12]. The number of lymph nodes that must be removed for pN classification remains controversial. Extensive dissection may increase the number of metastatic nodes and thus stage migration. Two analyses indicated that at least five lymph nodes should be removed from patients with duodenal tumors and nine from those with ileal and jejunal tumors [13,14]. In a report that evaluated both duodenal and jejonoileal tumors, eight lymph nodes were considered sufficient [10,11]. However, other data suggest that the evaluation of more lymph nodes may better predict the survival of SBA patients [15].

Data on adequate numbers of lymph nodes predict survival, as does the ratio of metastatic lymph nodes to the number of lymph nodes evaluated (the N ratio) [10,11,13,14,16]. Thus, both the total number of removed lymph nodes and the number of metastatic nodes predict the prognosis of SBA patients [10]. The N ratio is independently prognostic, with low N ratios predicting better survival [10,13]. We evaluated the prognostic significance of the number of lymph nodes removed from, and the N ratio of, SBA patients who underwent radical resection. We also determined the effects of various clinicopathological features on disease-free survival (DFS) and disease-specific survival (DSS), and identified potentially prognostic factors.

## 2. Patients and Methods

Between May 2001 and August 2020, data on 143 patients with adenocarcinomas of the duodenum, ileum, and jejunum treated in nine Turkish hospitals were retrospectively reviewed. Radically resected SBA patients of histologically confirmed R0 status (no residual microscopic or macroscopic tumors) and who survived for at least 3 months postoperatively were included. Patients of advanced stage and with secondary malignancies other than SBA were excluded. A total of 97 patients met the inclusion criteria. From patient files, we obtained age at diagnosis; gender; Eastern Cooperative Oncology Group (ECOG) performance score; resection type; tumor location; histopathological features and grade; lymphatic, vascular, and perineural invasion status; pathological T stage (pT); any lymph node involvement; TNM stage; resection margin; recurrence status; and survival. Staging followed the recommendations of the 2017 (8th edition) American Joint Committee on Cancer; we evaluated the clinical, radiological, and pathological findings at diagnosis [12].

### 2.1. Resected Lymph Node Number Cutoff

Receiver operating curve analysis yielded a cutoff of 13.40, which was very close to the median value (13) after outliers were trimmed. Thus, patients were divided into two groups, with <13 and ≥13 resected lymph nodes, respectively.

### 2.2. N Ratio Cutoff Values

The N ratio cutoff was based on a series of cutpoint analyses that employed Cox regression to maximize the chi-squared likelihood value. The maximum value was 0.28, which was the median after the trimming of outliers. Patients were divided into groups with N ratio (i.e., number of lymph node metastases divided by the number of removed lymph nodes) values of 0.00, 0.02–0.28, or >0.28; these were termed N ratio groups 0, 1, and 2, respectively.

## 3. Statistical Analyses

All statistical analyses were performed using MedCalc Software v19.7.2 (Ostend, Belgium; https://www.medcalc.org; (accessed on 1 January 2021)) and IBM SPSS Statistics for Windows v28.0 (Armonk, New York, NY, USA). The Shapiro–Wilk test was used to explore the normality of continuous variables. Descriptive statistics included means with standard deviations, and medians with ranges. Categorical variables are expressed as frequencies (*n*) with percentages (%). The Mann–Whitney U-test and Kruskal–Wallis test were used to compare non-normally distributed variables of two and more than two groups, respectively. The Bonferroni-adjusted Mann–Whitney U-test was used for *post hoc* comparisons. DSS and DFS analyses were performed using the Kaplan–Meier method. Median survival times were compared using the log-rank test. A Cox regression model was used for multivariate evaluation of factors affecting DSS. Statistical significance was determined at a level of *p* < 0.05.

## 4. Results

Data on 97 patients (41 female, 56 male) with radically resected SBAs were retrospectively analyzed. The median age at diagnosis was 58 years (range, 21–81 years); 55 (56.7%) were younger than 60 years. The most common tumor site was the duodenum (*n* = 57; 58.8%), and the tumor frequency decreased distally (jejunum: *n* = 24, 24.7%; ileum: *n* = 16, 16.5%). All patients with ileal and jejunal tumors underwent segmental resection; 31 patients with duodenal tumors underwent segmental resection, while the remaining 26 underwent pancreaticoduodenectomy. Most patients were of stage pT3 (*n* = 60; 61.9%). In terms of the pN classification based on lymph node metastases, 33 (34%) patients were pN0, 29 (29.9%) pN1, and 35 (36.1%) pN2; most (*n* = 64; 66%) were of TNM stage 3. The median number of dissected lymph nodes was 13 (range, 2–45) and the median number of lymph node metastases was 2 (range, 0–17). N ratio groups 0, 1, and 2 included 33 (34%), 29 (29.9%), and 33 (36.1%) patients, respectively. The N ratio correlated significantly with both the pN classification and TNM stage, and was significantly higher in patients exhibiting perineural and/or vascular invasion. Correlations between the N ratio and clinicopathological features are shown in Table 1.

The median follow-up time was 53 (range, 6–205) months; 54 (55.6%) patients died. The number of SBA-related deaths was 46 (47.4%). The 3- and 5-year DFS rates were 67.6 and 49.4% and the DSS rates were 75.8% and 59.5, respectively. By the TNM stage, the 5-year DFS rates of stage I, II, and III patients were 100%, 74.3%, and 31.8% and the 5-year DSS rates were 100%, 77.8%, and 47.2%, respectively. The median DFS and DSS were not attained by patients of stages I and II; the median DFS and DSS were lower in stage III patients; thus, 40 (standard error (SE), 8.5; 95% confidence interval (CI), 23.4–58.3) and 52 (SE, 7.8; 95% CI, 37.6–67.5) months, respectively. This difference was significant in terms of both DFS and DSS (both, *p* < 0.001). The DSS curve is shown in Figure 1.

According to the pN classification, the 5-year DFS rates were 80.5, 60.9, and 8.8% and the 5-year DSS rates were 83.2, 84.5, and 17.7% for the pN0, 1, and 2 groups, respectively. The median DFS and DSS were not attained by pN0 or pN1 patients and were lower in pN2 patients, at 18 (SE, 6.4; 95% CI, 13.3–23.2) and 31 (SE, 6.2; 95% CI, 28.6–52.9) months, respectively. The differences were significant in terms of both DFS and DSS (both, *p* < 0.001). The DSS curve is shown in Figure 2. The 5-year DFS rates of N ratio groups 0, 1, and 2 were 80.5%, 72%, and 2.8%; the 5-year DSS rates were 95.7%, 83.2%, and 13.9%, respectively. The median DFS and DSS were attained by N ratio groups 0 and 1; median DFS and DSS were lower in N ratio group 2, at 18 (SE, 3.5; 95% CI, 17–31) and 30 (SE, 3.8; 95% CI, 96.3–132.9) months, respectively. This difference was significant in terms of both DFS and DSS (both, *p* < 0.001). The DSS curve is shown in Figure 3.

More than half of all patients (52.4%) underwent dissection of ≥13 lymph nodes. In those for whom <13 and ≥13 lymph nodes were dissected, the 5-year DFS rates were 42.4% and 55.7% and the 5-year DSS rates were 47.7% and 70.6%, respectively. The median DFS and DSS were not attained in the latter group and were 42 (SE, 13.5; 95% CI, 74.2–127.1) and 60 (SE, 7.6; 95% CI, 45.1–74.9) months, respectively, in the former group (both, *p* = 0.075). The DSS curve is shown in Figure 4. Although the numerical survival was better in the latter group, there was no significant between-group difference in terms of DFS or DSS.

In terms of primary tumor resection, 31 patients with duodenal tumors underwent segmental resection, and 26 underwent pancreoticoduodenectomy. The median numbers of lymph nodes removed during these operations were 12 and 13, respectively, which were not significantly different (*p* = 0.602). In the latter and former patients, the 5-year DFS rates were 46.1% and 42.7% and the 5-year DSS rates were 58.7% and 60%, respectively (*p* = 0.760 and *p* = 0.660, respectively), with no significant difference.

The univariate analysis of clinicopathological factors showed that vascular and perineural invasion, N stage, TNM stage at diagnosis, and the N ratio significantly affected DSS. The N ratio was closely associated with DSS after radical SBA resection (Table 2). Significant variables (*p* < 0.05) according to univariate analysis were included in the multivariate model. The N ratio, TNM stage, and N stage were closely associated. Therefore, we used a Cox proportional hazard model to perform multivariate analysis. The results revealed that the N ratio (chi-squared, 12.8; *p* < 0.001; hazard ratio (HR), 9.75; 95% CI, 2.72–34.82) and perineural invasion (chi-squared, 7.53; *p* = 0.006; HR, 4.21; 95% CI, 1.50–11.78) were independently prognostic (Table 3).

## 5. Discussion

The primary treatment for local SBA is surgical resection and *en bloc* lymph node removal. Regional lymph nodes at risk of metastasis must be removed [3,17]. In patients lacking distant metastases, regional lymph node metastasis is an important predictor of oncological outcomes [11,18,19]. Lymph node metastasis is a major consideration when a clinician chooses adjuvant chemotherapy to improve prognosis. In patients with colorectal and gastric cancers, the numbers of lymph nodes that should be removed to improve prognosis and ensure accurate staging are well-defined [20,21,22,23]; this is not the case for SBAs. Approximately half of our patients (52.4%) underwent dissection of ≥13 lymph nodes. The 5-year DSS rate was numerically better in this group than in the group for whom fewer nodes were removed (47.7% vs. 70.6%); however, statistical significance was not attained (*p* = 0.075), perhaps due to the small sample size.

In a retrospective analysis of 1091 non-metastatic SBA patients whose information was entered into the Surveillance, Epidemiology, and Results (SEER) database from 2004 to 2011, the removal of at least nine lymph nodes was associated with better overall survival and cancer-specific survival [14]. Similarly, a retrospective analysis of 1991 SBA patients whose information was entered into the SEER registry from 1998 to 2007 showed that the removal of at least eight lymph nodes improved both stage I/II and stage III cancer-specific survival [11]. One study found that the evaluation of more lymph nodes might better predict the survival of SBA patients [15].

No prospective study has yet explored the relationship between the surgical technique used and the number of lymph nodes removed; all data are retrospective [7,24,25]. The type of resection usually depends on the location of the primary tumor. Although segmental resection is the mainstay, pancreotiduodenectomy may be required by patients with duodenal tumors. In our study, there were 57 such patients, of whom 26 underwent pancreotidoduodenectomy and 31 underwent segmental resection. All patients with SBAs in the ileum and jejunum underwent segmental resection. There was no significant between-technique difference in terms of either the number of lymph nodes removed or the DSS. Similarly, a meta-analysis of 6438 duodenal cancer patients reported that both techniques allowed adequate lymph node dissection when tumors lay in the distal duodenum; overall survival did not differ significantly [24]. Two retrospective analyses came to the same conclusions [25,26]. Limited resection reduces morbidity and postoperative fistulation [26] and may therefore be appropriate for selected patients with tumors in duodenal segments 3–4, although further studies are required.

The number of involved lymph nodes is a very important predictor of prognosis in SBA patients. Regardless of T stage, lymph node involvement affects survival outcomes. In our study, the best DSS rate was that of the pN0 group, and the worst that of the pN2 (≥3 lymph node metastases) group. In a retrospective analysis, patients with lymph node involvement were divided into those with <3 and ≥3 metastatic nodes. Cancer-specific survival was better in the former patients [11], which is consistent with our results. Indeed, in the 2017 (8th edition) American Joint Committee on Cancer, stage 3 tumors were classified by the number of involved lymph nodes (IIIA < 3; IIIB ≥ 3 lymph nodes) [12].

Several analyses and retrospective studies of SBA patients whose data are in the SEER database have sought associations between the numbers of lymph nodes removed and a low N ratio on survival after surgery [10,11,13,14,16]. The N ratio assesses both the total number of lymph nodes removed and the number of involved nodes on the same scale when predicting prognosis. However, no consensus N ratio cutoff for SBA patients has emerged. We found inverse relationships between the N ratio and both DFS and DSS. The best DFS and DSS rates were those of the N ratio 0 group (0.00) and the worst were those of the N ratio 2 group (>0.28); the difference was significant. Multivariate analysis also revealed that the N ratio was independently prognostic of DSS.

Similarly, in a retrospective analysis of SBA patients (*n* = 1991) whose information was entered into the SEER database from 1998 to 2005, the survival of stage III SBA patients with N ratios of 0.02–0.20, 0.21–0.50, or 0.52–1 were stratified; survival decreased as the N ratio increased. The survival of the N ratio 0.52–1 group was significantly poorer than those of the other two groups [11]. In a retrospective analysis of 2772 SBA patients registered in the SEER database between 1988 and 2010, survival decreased as the N ratio increased, and the poorest median survival (16 months) was that of a group with N ratio > 0.4 [13]. Another study published in 2022 year was used an N ratio cutoff of 0.4, and found that the N ratio, the number of lymph nodes resected, and the number of positive lymph nodes enhanced prognostic accuracy [27].

Unfortunately, no consensus N ratio cutoff has yet emerged; all data are based on retrospective analyses. SBA is heterogeneous in terms of the tumor location, optimal surgical technique, grade, and stage. The risk of recurrence is high, especially in patients lacking metastases but exhibiting lymph node involvement (stage III). Adjuvant chemotherapy improves prognosis. No prospective study has yet evaluated whether adjuvant chemotherapy improves survival after SBA removal; again, all data were retrospectively derived. Clinicians who choose adjuvant treatments consider the few retrospective studies on SBAs and prospective studies on colorectal adenocarcinomas [6,16,28,29,30]. A better understanding of the factors predicting prognosis, especially of stage III SBAs, might identify patients that would particularly benefit from adjuvant treatment. More aggressive adjuvant treatments may be appropriate for patients at high risk of recurrence.

Our work had certain limitations. Any retrospective evaluation is associated with a risk of selection bias. Although we combined the experience of multiple Turkish institutions, given the rarity of SBAs, our moderate sample size is another limitation of our study.

## 6. Conclusions

We found that the N ratio was independently prognostic of DSS in patients with radically resected SBAs. Such patients require appropriate lymphadenectomy. Further prospective studies are required to explore the prospective utility of the N ratio as a prognostic factor.

## Figures and Tables

**Figure 1 medicina-59-01472-f001:**
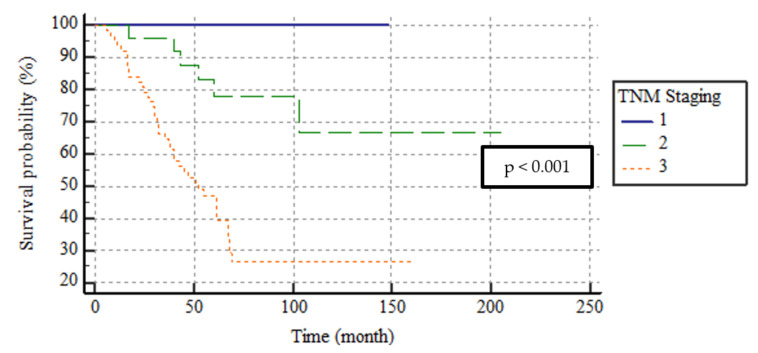
The survival curves of DSS according to TNM stage.

**Figure 2 medicina-59-01472-f002:**
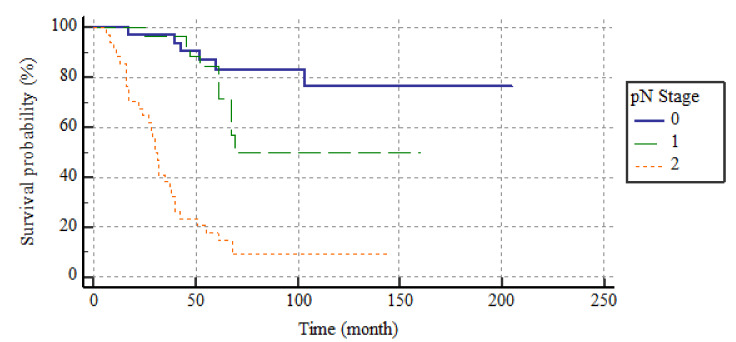
The survival curves of DSS according to pN stage.

**Figure 3 medicina-59-01472-f003:**
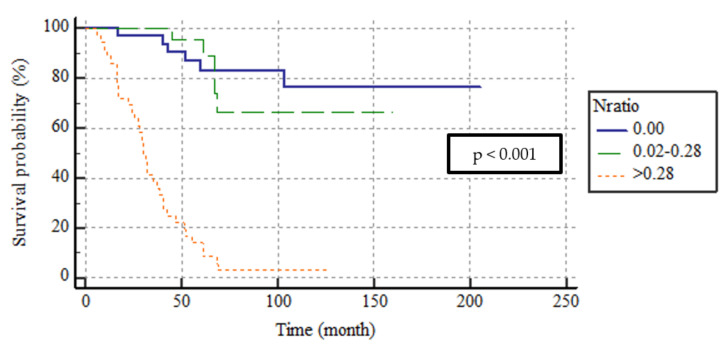
The survival curves of DSS according to N ratio categorization.

**Figure 4 medicina-59-01472-f004:**
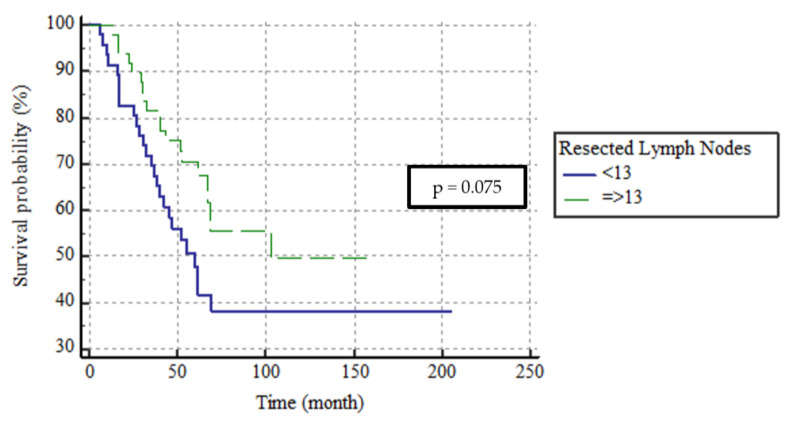
The survival curves of DSS according to removed lymph node categorization.

**Table 1 medicina-59-01472-t001:** The relation between the N ratio and clinicopathological characteristics.

Variable	Med (IQR)	*p*
**Gender**		0.800 ^1^
*Female n = 41*	16.6 (0–43.5)	
*Male n = 56*	8.3 (0–38.2)	
**Age at diagnosis (in categories)**		0.476 ^1^
*<60 n = 55*	8.3 (0–38.4)	
*≥60 n = 42*	13.7 (0–42.1)	
**Histopathology**		NA
*Adenocarcinoma n = 87*	8.3 (0–38.9)	
*Signet ring cell n = 7*	33.3 (0–42.8)	
*Mucinous n = 3*	25 (na)	
**Tumor Differentiation**		0.686 ^2^
*Well differentiated n = 21*	11.1 (0–45.1)	
*Moderately differentiated n = 56*	7.7 (0.34.4)	
*Poorly differentiated n = 14*	32.4 (6.1–62.3)	
**TNM Stage**		**<0.001 ^2^**
*Stage I n = 8*	0 (0–0)	
*Stage II n = 25*	0 (0–0)	
*Stage III n = 64*	32.4 (8.8–46.5)	
**pN Stage**		**<0.001 ^2^**
*pN0 n = 33*	0 (0–0)	
*pN1 n = 29*	8.3 (6.9–18.2)	
*pN2 n = 35*	44 (37.5–62.5)	
**pT Stage**		0.119 ^2^
*1 n = 2*	0 (0–0)	
*2 n = 10*	0 (0–23.6)	
*3 n = 60*	8.9 (0–44.1)	
*4 n = 25*	16.6 (0–42.7)	
**Vascular Invasion**		**<0.001 ^1^**
*(−) n = 43*	0 (0–0)	
*(+) n = 53*	36.3 (8.7–47)	
**Perineural Invasion**		**<0.001 ^1^**
*(−) n = 39*	0 (0–8.3)	
*(+) n = 58*	33.3 (7.6–46.2)	
**Lymph nodes Invasion**		**<0.001 ^1^**
*(−) n = 33*	0 (0–0)	
*(+) n = 64*	32.4 (8.8–46.5)	
**Category of resected lymph nodes**		0.432 ^1^
*<13 n = 46*	17.4 (0–46.6)	
*≥13 n = 51*	7.7 (0–38.4)	

^1^ Mann–Whitney u test, ^2^ Kruskal–Wallis test.

**Table 2 medicina-59-01472-t002:** Univariate analysis according to clinicopathological factors and disease-specific survival.

Factor		No. of Patients (%)	Median DSS Time (Months)	95% CI	*p*
**All**		**97**	68	25–111	-
**Gender**					0.145
** *Male* **		41 (42.3)	125.2	101.6–148.8	
** *Female* **		56 (57.7)	78.3	60.5–96.3	
**Age (year)**					0.883
** *<60* **		55 (56.7)	69	22.4–115.6	
** *≥60* **		42 (43.3)	68	-	
**Tumor Location**					0.667
** *Duodenum* **		57 (58.8)	109.4	86.7–132.2	
** *Jejenum* **		24 (24.7)	98.8	75.3–122.2	
** *Ileum* **		16 (16.5)	91.0	59.9–122.2	
**Surgery Type**					0.998
** *Segmental resection* **		71 (73.2)	69	23.6–114.4	
** *Pancreaticoduodenectomy* **		26 (26.8)	67	-	
**Histopathology**					NA
** *Adenocarcinoma* **		87 (89.7)	68	25.5–110.5	
** *Signet ring cell* **		7 (7.2)	47	21.3–72.7	
** *Mucinous* **		3 (3.1)	-	-	
**Tumor Differentiation**					0.331
** *Well differentiated* **		21 (23.1)	78.4	54–102.9	
** *Moderately differentiated* **		56 (61.5)	128	103.7–152.3	
** *Poorly differentiated* **		14 (15.4)	86	52.6–119.4	
**Vascular Invasion**					**<0.001**
** *Present* **		53 (55.2)	72.0	51.7–92.3	
** *Absent* **		43 (44.8)	128.5	115.3–141.7	
**Perineural Invasion**					**<0.001**
** *Present* **		58 (59.8)	67.1	52.1–82	
** *Absent* **		39 (40.2)	175.0	153.2–197	
**pT Stage**					0.133
** *1* **		2 (2.1)	85	85–85	
** *2* **		10 (10.3)	115.2	83.6–146.9	
** *3* **		60 (61.9)	116.4	93.2–139.6	
** *4* **		25 (25.8)	70.4	45.5–95.3	
**pN Stage**					**<0.001**
** *pN0* **		33 (34)	170.9	146.4–195.4	
** *pN1* **		29 (29.9)	109.2	84.4–133.9	
** *pN2* **		35 (36.1)	40.8	28.6–52.9	
**TNM Stage**					**<0.001**
** *Stage I* **		8 (8.2)	148	148–148	
** *Stage II* **		25 (25.8)	157.9	125.2–190.6	
** *Stage III* **		64 (66)	71.5	56.1–86.9	
**N Ratio**					**<0.001**
** *N ratio 0* **	** *(0.00)* **	33 (34)	170.8	146.3–195.4	
** *N ratio 1* **	** *(0.02–0.28)* **	29 (29.9)	127.7	104.1–151.2	
** *N ratio 2* **	** *(>0.28)* **	35 (36.1)	35.1	27.6–42.6	
**Resected lymph nodes**					0.075
** *<13* **		46 (47.4)	60	45.1–74.9	
** *≥13* **		51 (52.6)	103	-	

**Table 3 medicina-59-01472-t003:** Multivariate analyses of association among covariates and DSS.

Factors	Wald	*p*	HR	95% CI
**Vascular invasion**	0.103	0.748	0.812	0.23–2.89
**Perineural Invasion**	**7.534**	**0.006**	**4.21**	**1.50–11.78**
**N ratio**	27.54	<0.001		
** *0.00 vs. 0.02–0.28* **	0.381	0.537	0.664	0.18–2.44
** *0.00 vs. >0.28* **	**12.28**	**<0.001**	**9.75**	**2.72–34.82**

## Data Availability

Not applicable.

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
