# Peer review of "The Prognostic Utility of the Metastatic Lymph Node Ratio and the Number of Regional Lymph Nodes Removed from Patients with Small Bowel Adenocarcinomas"

_medicina, 2023, doi:10.3390/medicina59081472_

Round 1

Reviewer 1 Report

The authors have explained the prognostic significance of the number of removed lymph nodes and N ratio in radically resected SBA patients.  I would like the authors to add figures and tables of their statistical analysis for better understanding of the work.

kindly check the grammatical errors in the paper.

Author Response

I would like to thank for  your valuable criticisms.The response to comments are indicated in below part.

Peer Reviewer:

Comments to the Author

1-The authors have explained the prognostic significance of the number of removed lymph nodes and N ratio in radically resected SBA patients.  I would like the authors to add figures and tables of their statistical analysis for better understanding of the work.

The figures and tables were added to the main manuscript.

2- Kindly check the grammatical errors in the paper.

Grammatical errors in the parer were corrected.

The English in this document has been checked by at least two professional editors, both native speakers of English. For a certificate, please see: http://www.textcheck.com/certificate/rBdK16

Reviewer 2 Report

the author wanted to report on their research foEvaluation of the prognostic value of metastatic lymph node ra- tio and number of regional lymph nodes removed in small bowel adenocarcinomas 

Please correct the language according to the standard writing standards in English and according to the rules of the journal.

This research is  good, in presentation research what do you need to add to your take home massage in research 

it would be better if this review also contains pictures related to the topic

the conclusions contained in the manuscript are too long and do not focus on this issue, please edited

Manuscripts need a lot of improvement to be published in this journal.

Please correct the language according to the standard writing standards in English and according to the rules of the journal.

Author Response

I would like to thank for  your valuable criticisms.The response to comments are indicated in below part.

Peer Reviewer:

Comments to the Author

1- Please correct the language according to the standard writing standards in English and according to the rules of the journal.

Grammatical errors in the paper were correcte. The English in this document has been checked by at least two professional editors, both native speakers of English.For a certificate, please see : http://www.textcheck.com/certificate/rBdK16

2- This research is  good, in presentation research what do you need to add to your take home massage in research ? 

In the conclusion section, we hava changed as ‘’ We found that the N ratio was independently prognostic of DSS in patients with radically resected SBAs. Such patients require appropriate lymphadenectomy. Further prospective studies are required to explore the prospective utility of the N ratio as a prognostic factor.’’.

3- it would be better if this review also contains pictures related to the topic

The figures and tables were added to the main manuscript.

4- the conclusions contained in the manuscript are too long and do not focus on this issue, please edited

In the conclusion section, we hava changed as ‘We found that the N ratio was independently prognostic of DSS in patients with radically resected SBAs. Such patients require appropriate lymphadenectomy. Further prospective studies are required to explore the prospective utility of the N ratio as a prognostic factor.’’

5- Manuscripts need a lot of improvement to be published in this journal.

The figures and tables were added to the main manuscript. We have also corrected some grammatical errors. As you requested, we hava changed  the section of conclusion.

Reviewer 3 Report

Authors aimed to define clinicopathologic features, to determine the effect of clinicopathologic features on disease free survival (DFS) and disease spesific survival (DSS), and to identify potential prognostic factors.

Small bowel adenocarcinoma (SBA) is an uncommon cancer of the gastrointestinal tract. Its mechanisms are poorly understood. Due to the lack of large-scale, multicenter, randomized controlled trials, the optimal therapeutics are controversial. Since it is rare and seen in advanced stages, its treatment is difficult and important. Therefore, studies on this cancer are important.

Plagiarism checked, it is suitable for ethical situaiton

Comments;

 Abstract section: Use the long form of the abbreviation where DSS and DFS are first mentioned in the summary section, and the short form when the second is mentioned.

Conclusions section: In this article, it is emphasized that there are need prospective studies. This emphasis can be added to the conclusions section.

References: References partially dated, current references can be used.

I hope my comments will contribute to your article, good lucks to authors.

Author Response

I would like to thank for  your valuable criticisms.The response to comments are indicated in below part.

Peer Reviewer:

Comments to the Author

1.Use the long form of the abbreviation where DSS and DFS are first mentioned in the summary section, and the short form when the second is mentioned.

The long form of the abbreviation where DSS and DFS are first mentioned in the summary section was used.

  1. Conclusions section: In this article, it is emphasized that there are need prospective studies. This emphasis can be added to the conclusions section.

In the conclusion section, this was empsasised as an ‘’ We found that the N ratio was independently prognostic of DSS in patients with radically resected SBAs. Such patients require appropriate lymphadenectomy. Further prospective studies are required to explore the prospective utility of the N ratio as a prognostic factor.’’

  1. References: References partially dated, current references can be used.

Although many studies are retrospective, we could not find many studies about N ratio in recently. We added a study published in 2022 to the conclusion part.

Another study published in 2022 year was used an N ratio cutoff of 0.4, and found that the N ratio, the number of lymph nodes resected, and the number of positive lymph nodes enhanced prognostic accuracy [27].
